# Aligning Signal Leakage matters for Synthetic Data Generation of Satellite Imagery

## Abstract

While satellite data is essential for applying computer vision to many real-world tasks, it remains expensive to acquire. Although other computer vision tasks have alleviated data procurement costs by augmenting training datasets with synthetic images from text-to-image models, such augmentation remains underdeveloped in the remote sensing domain. In this work, we propose an alternative approach for generating synthetic training data tailored to satellite imagery. To better understand the underlying problem, we begin by analyzing the impact of the target data distribution in comparison to the distributions used to train the text-to-image generation model. We find that *data rarity* is strongly correlated with the effectiveness of synthetic training data produced by Stable Diffusion fine-tuned on few-shot examples, suggesting that rarity can serve as a low-cost proxy for pre-evaluating the effectiveness of synthetic data generation. Notably, our analysis shows that Stable Diffusion struggles to produce useful training images for rare, out-of-distribution data. Building on this insight, we propose two modifications to the generation process tailored to satellite images: *offset noise* and *leak-aligned noise*. Both are designed to adjust the initial noise distribution and correct low-frequency characteristics. Our approaches enable improved training performance for classifiers trained on synthetic data, demonstrated on three satellite benchmarks.

## 1 Introduction

Remote sensing imagery is crucial in leveraging computer vision for many real-world applications, such as environmental monitoring, urban development, agricultural management, and wildlife conservation. However, modern vision models are data-hungry, often requiring datasets with millions or even billions of images. One of the most widely used open-source datasets, LAION-5B (Schuhmann et al., 2022), has over 5 billion images. Unfortunately, acquiring satellite images can be complicated and costly given the cutting-edge technology required. Such limitations are especially detrimental in niche or local tasks that cannot leverage existing large-scale open-source datasets. To address data scarcity, other computer vision tasks may augment existing datasets with synthetic images from increasingly powerful text-to-image models like FLUX (Labs, 2024) and SD3 (Esser et al., 2024). For instance, DataDream (Kim et al., 2024) uses few-shot data to finetune the pre-trained diffusion model, and generate synthetic data for downstream classification training. While DataDream achieves $13.0\%$ improvement on FGVC Aircraft (Maji et al., 2013a) compared to training with real few-shot data alone, it does not always generalize to different benchmarks; DataDream does not improve over simple fine-tuning on real data when evaluated on EuroSAT (Helber et al., 2019), an earth observation dataset. This raises the question: under what conditions does DataDream fail to provide benefits, and what underlying factors drive this limitation?

To investigate this limitation, we examine the distributional shift between downstream benchmarks and the pretraining dataset of the diffusion model (i.e. LAION). We hypothesize that when the downstream distribution diverges significantly from the pretraining distribution, fine-tuning the diffusion model becomes less effective at adapting to the new domain. To quantify this, we employ rarity score (Han et al., 2022), which measures how uncommon the downstream distribution is relative to the pretraining distribution. Specifically, for each downstream benchmark, we compute the accuracy gap between two models: one trained with synthetic data generated by DataDream and the other with few-shot real data (reported as Real-acc – DataDream-acc). We then analyze the correlation between this performance gap and the rarity score. As shown in Figure 1 (a), we observe a strong

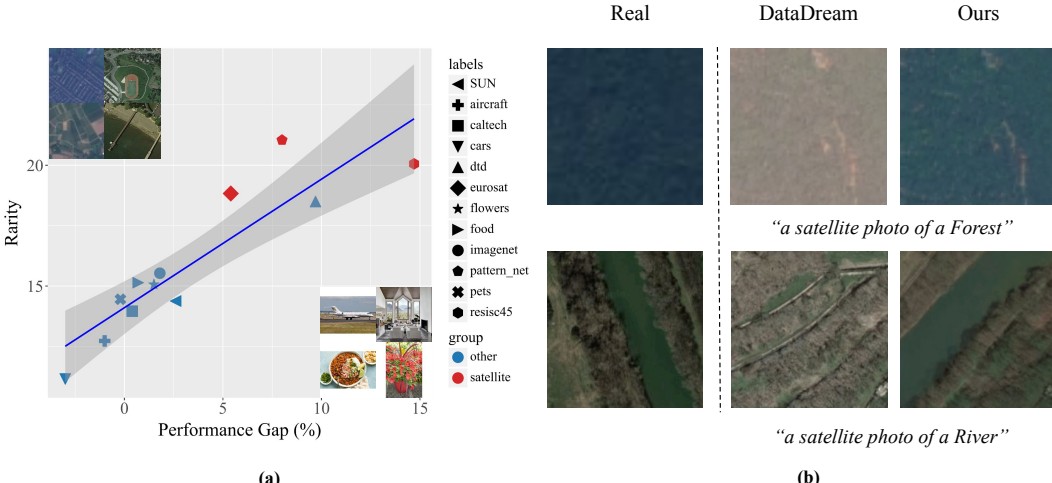

Figure 1: (a) Synthetic training images boost performance less when target data is out-of-distribution (e.g., satellite samples in the upper left) for the generative model (trained with LAION-data shown in the lower right). Performance gap refers to the Top-1 accuracy gap between classifiers trained on real versus synthetic data from DataDream (Kim et al., 2024). Rarity measures the distribution shift of the listed datasets to LAION, higher rarity indicates larger distribution shift. (b) Our method corrects low-frequency cues (e.g., color) for higher-quality training data.

positive correlation, indicating that DataDream underperforms more severely on benchmarks that are rarer with respect to the pretraining distribution. For datasets like EuroSAT (Helber et al., 2019) and DTD (Cimpoi et al., 2014), which differ substantially from LAION's distribution, fine-tuning is less effective at adapting the model to the target classes.

For satellite datasets far from the diffusion model's training distribution, we propose an alternative, it's more effective in satellite image generation than the SOTA DataDream (Kim et al., 2024), as shown in Figure 1(b). Recent work shows that standard diffusion noise schedules leak image information, particularly in the low-frequency domain (e.g., color), leading models to rely on this leakage during inference (Lin et al., 2024). This leakage may help in-distribution classes but becomes problematic with greater distribution shift. We introduce two noise adaptation strategies tailored for out-of-distribution datasets like satellite data: offset and leak-aligned noise. Offset noise adjusts the standard deviation of the added noise to better match the low-frequency characteristics of the target dataset, helping approximate more appropriate inductive biases. In contrast, leak-aligned noise takes a more principled approach: it estimates the noise aligned with the leakage observed during DataDream fine-tuning, allowing for a more dataset-specific and class-specific noise calibration.

We make the following contributions: 1) We analyze failure cases of existing few-shot methods for generating synthetic training data, finding that *rarity* is strongly correlated with synthetic training performance; 2) We introduce *offset noise* and *leak-aligned noise* as targeted solutions for handling satellite data and other out-of-distribution datasets; and 3) We evaluate the impact of these noise adaptations on downstream model performance when trained with synthetic satellite data.

## 2 RELATED WORKS

Synthetic training images have garnered growing interest, driven by recent advances in image generation models. In the following sections, we review key developments in these areas.

**Training with Synthetic Data.** Synthetic images have gained attention as training data, with studies exploring augmenting real datasets with synthetic images (Bansal & Grover, 2023; Burg et al., 2023; Dunlap et al., 2023; Zhou et al., 2023) or pre-training models on synthetic data followed by few-shot fine-tuning on real data (Hammoud et al., 2024; Tian et al., 2024). Other research has investigated training exclusively on synthetic datasets (Hammoud et al., 2024; Sariyildiz et al., 2022). Efforts generally center on two key properties: *faithfulness* and *diversity*. For faithfulness, there are

techniques like filtering (Dunlap et al., 2023; He et al., 2023; Lin et al., 2023) and including class information (Sariyildiz et al., 2022). Diversity can be enhanced by reducing guidance scale (Sariyildiz et al., 2022), generating varied prompts with language models (Hammoud et al., 2024; He et al., 2023), incorporating domain or background details (Sariyildiz et al., 2022; Dunlap et al., 2023; Shipard et al., 2023), or using multiple prompt templates (Burg et al., 2023).

Faithfulness also improves by using a few real images. Three few-shot approaches stand out. IsSynth (He et al., 2023) seeds generation with partially-noised real images and filters out low-similarity outputs. DISEF (da Costa et al., 2023) similarly seeds generation but adds diversity using prompts from real image captions. The state-of-the-art (SOTA) DataDream (Kim et al., 2024) fine-tunes the diffusion model via LoRA (Hu et al., 2021) on real data to improve class understanding.

Synthetic training data is well-studied for general datasets but underexplored in satellite imagery, whose distinct distribution may limit method transferability. Nguyen et al. (2024) fine-tuned Stable Diffusion on satellite data with DreamBooth (Ruiz et al., 2022) and Textual Inversion (Gal et al., 2023), evaluating similarity to real images. Nguyen et al. (2024) investigated rare objects in satellite imagery. Both showed synthetic satellite imagery is feasible but left ample room for improvement.

**Synthetic Image Generation.** Synthetic data quality relies on the generative model, with recent advances achieving photorealism; earlier methods were dominated by Variational Autoencoders (Kingma et al., 2013) or Generative Adversarial Networks (Goodfellow et al.) which provided key groundwork but had limited fidelity. Recently, the field shifted to Latent Diffusion and Flow-Matching Models, boosting image quality and controllability (Rombach et al., 2022). SOTA models like SD2.1 (Rombach et al., 2022), SDXL (Podell et al., 2023), FLUX (Labs, 2024), DALL-E 3 (Betker et al., 2023), Imagen 2 (Saharia et al., 2022), and Midjourney v6 (Midjourney, 2024) exemplify this new generation of high-performing architectures.

**Initial Noise Selection.** Though theory treats all noise inputs equally, initial noise choice significantly affects image quality in practice (Eyring et al., 2024; Everaert et al., 2023; Qi et al., 2024; Eyring et al., 2025). Moreover, diffusion models can retain low-frequency information (e.g. color) in the fully noised latent (Lin et al., 2024). This literature drives our decision to investigate the effects of initial latent noise on satellite data.

# 3 ANALYSIS

First, we examine how the target data distribution affects synthetic data quality to explain synthetic satellite dataset underperformance. We empirically compare the distribution of downstream training datasets to LAION (Schuhmann et al., 2022) (the training dataset for our chosen diffusion model) and analyze the relation to performance gains achieved using synthetic images (Kim et al., 2024). We then explore the performance failure theoretically to propose a better solution for satellite data.

## 3.1 PRELIMINARY

The analysis is inspired by a limitation noted in DataDream (Kim et al., 2024). DataDream effectively generates images with fine-grained details and key objects, enhancing the quality of synthetic datasets where such features are crucial for class discrimination (e.g., aircraft). Training with these improved synthetic data further enhances the classification performance of downstream models. However, its effectiveness depends on the target data distribution and its similarity to generative model training data distribution (i.e., LAION for Stable Diffusion). When comparing the classification performance of models trained with synthetic data only and few-shot real images only in DataDream, the performance gap is significantly larger for datasets DTD (Cimpoi et al., 2014) and EuroSAT (Helber et al., 2019), while synthetic data from other datasets can achieve comparable performance as real data in classifier training. We hypothesize that the more pronounced performance gap can correlate to the large distribution shift of DTD and EuroSAT to LAION compared to other downstream datasets.

## 3.2 IMPACT OF DATA DISTRIBUTIONS

To investigate the relationship between classification performance gap and the distribution shift across the target and training datasets, it is crucial to find a metric that accurately represents the

Table 1: Data distribution comparison between downstream training datasets and LAION. Downstream datasets: ImageNet (IN) (Russakovsky et al., 2015), Caltech 101 (CAL) (Li et al., 2022), DTD (Cimpoi et al., 2014), EuroSAT (EuSAT) (Helber et al., 2019), FGVC Aircraft (AirC) (Maji et al., 2013b), Oxford Pets (Pets) (Parkhi et al., 2012), Stanford Cars (Cars) (Krause et al., 2013), SUN397(SUN) (Xiao et al., 2010), Food101 (Food) (Bossard et al., 2014), Flowers 102 (FLO) (Nilsback & Zisserman, 2008), PatternNet (Patt) (Zhou et al., 2018), NWPU-RESISC45 (RES) (Cheng et al., 2017). LAION includes 411 shards from LAION-400m (Schuhmann et al., 2021).

| metric | IN | CAL | DTD | EuSAT | AirC | Pets | Cars | SUN | Food | FLO | Patt | RES |
|---|---|---|---|---|---|---|---|---|---|---|---|---|
| density | 0.38 | 0.41 | 0.36 | **0.07** | 0.25 | 0.56 | 0.49 | 0.48 | 0.52 | 0.76 | **0.03** | **0.07** |
| rarity | 15.53 | 13.96 | **18.50** | **18.83** | 12.73 | 14.45 | 11.15 | 14.38 | 15.14 | 15.06 | **21.04** | **20.06** |

data distribution and compute the distribution shift. Because it is intractable to compare data distributions in pixel-space, we conduct and compare data distributions of downstream datasets and LAION in embedding space. We use the 10 datasets evaluated in DataDream, supplemented by two additional satellite datasets: PatternNet (Zhou et al., 2018) and RESISC45 (Cheng et al., 2017). We choose DINOv2 (Oquab et al., 2023) for feature extraction, as it considers the overall structure of the image while still effectively identifying key objects in the image, giving a faithful representation of data distributions. To quantitatively measure the difference between the distributions of the downstream training datasets and LAION, we use two metrics: density (Naeem et al., 2020) and rarity (Han et al., 2022). Overall, both metrics measure the overlap of pairwise data distributions. Specifically, they are based on defining k-NN (k-Nearest Neighbors) spheres with samples in one distribution and computing distances from samples in another distribution to these spheres. More specifically, density computes the number of k-NN spheres defined by LAION samples that contains at least one sample from another distribution. And rarity computes the minimum radius of LAION k-NN spheres that contains at least one sample from another distribution. Thus, in general low density and high rarity indicates a large domain shift to LAION.

In Table 1, we observe that all three satellite datasets exhibit significantly lower density and higher rarity than other datasets, indicating a larger distribution shift to LAION. DTD also presents a significantly higher rarity while the density is similar as non-satellite datasets. This observation suggests that rarity serves as a more reliable metric for computing the distribution shift in our scenario.

As illustrated in Figure 1 (a), we further confirm a significant linear correlation between distribution shift (represented by metric rarity) and classification performance gap, supported by the statistics Pearson coefficient (Sedgwick, 2012) $R = 0.90$ and p-value $= 7.14e - 5$. We also observe a significant linear correlation between density and performance gap, with details provided in Appendix A.3. This observation reveals that a large distribution shift hinders the model's ability to adapt to the target real-world domain, and that satellite data involves a particularly significant distribution shift. This further reduces the fidelity of synthetic images generated by the model, so that the classifier tuned on synthetic data can not achieve comparable performance as when tuned on real data.

Motivated by this analysis, we investigate how distribution shift impacts few-shot fine-tuning and synthetic data classification, focusing on its effect on the diffusion process.

### 3.3 MISALIGNED NOISE DISTRIBUTION BETWEEN TRAINING AND INFERENCE

Given a real sample $\boldsymbol{x} \sim q(\boldsymbol{x})$, diffusion training consists of a forward process that gradually adds Gaussian noise to a real image, and a reverse process that learns to denoise and reconstruct the image (Ho et al., 2020). With pre-defined noise schedule $\beta_1, ..., \beta_T$ and the reparameterization trick (Kingma et al., 2013), the noisy image at timestep t during forward process is represented by:

$$\boldsymbol{x}_t = \sqrt{\bar{\alpha}_t}\boldsymbol{x}_0 + \sqrt{1 - \bar{\alpha}_t}\boldsymbol{\epsilon}_0, \quad \boldsymbol{\epsilon}_0 \sim \mathcal{N}(\mathbf{0}, \mathbf{I}) \tag{1}$$

where $\alpha_t = 1 - \beta_t$, $\bar{\alpha}_t = \prod_{s=1}^{t} \alpha_s$ and $\boldsymbol{x}_0$ is the latent representation of sample $\boldsymbol{x}$. Theoretically, $\beta_t$ and thereby $\bar{\alpha}_t$ is chosen such that at the last diffusion step $T$, $\boldsymbol{x}_T$ is a pure Gaussian noise without any information about the image $\boldsymbol{x}_0$:

$$\boldsymbol{x}_T = \sqrt{\bar{\alpha}_T}\boldsymbol{x}_0 + \sqrt{1 - \bar{\alpha}_T}\boldsymbol{\epsilon}_0, \quad \boldsymbol{\epsilon}_0 \sim \mathcal{N}(\mathbf{0}, \mathbf{I}) \tag{2}$$

which means $\bar{\alpha}_T$ should reach zero. However, in practice, noise schedules are usually defined to avoid $\bar{\alpha}_T = 0$ to prevent division-by-zero issues during model training. The actual values of $\bar{\alpha}_T$ in

some common noise schedules have been measured and reported in Lin et al. (2024). The non-zero $\bar{\alpha}_T$ allows a signal leakage $\sqrt{\bar{\alpha}_T}x_0$ in $x_T$ and prevent it from being pure Gaussian noise. Moreover, the leaked signal is reported to contain low-frequency information (channel-mean) about $x_0$ (Lin et al., 2024). Since the reverse process is trained based on $x_T$, the leaked signal with channel-mean information enables the model to learn image reconstruction relying on the leaked signal, rather than learning the reconstruction from pure noise. However, in the inference phase, we feed a pure Gaussian noise $\epsilon_0 \sim \mathcal{N}(\mathbf{0}, \mathbf{I})$ as initial noise to the trained model for denoising and generating synthetic samples from the learned data distribution. Since the trained model relies on the leaked signal and fails to learn how to reconstruct the channel-mean information from pure noise, we get misrepresented low-frequency contents (e.g. color) in the generated images, which is especially problematic for satellite data. Compared to high-resolution web images, satellite images are usually in significantly lower resolution and can not be easily differentiated based on texture or fine-grained details. Instead, color becomes a important cue for distinguishing between classes. Thus, generating images that capture correct color of satellite images is crucial for the success of classification.

Why this effect is pronounced in domains with a large distributional gap from LAION (e.g., satellite imagery) but negligible in closer domains remains unclear. The key lies in the KL-divergence term in the VAE loss (Kingma et al., 2013), which drives the latent distribution of encoded data to approximate the standard Gaussian distribution $\mathcal{N}(\mathbf{0}, \mathbf{I})$. Due to pretraining on LAION, Stable Diffusion's VAE maps LAION-like data to a standard Gaussian in latent space. Hence, even if there is a leaked signal $\sqrt{\bar{\alpha}_T}x_0$ remaining in $x_T$ during training, $x_T$ still follows $\mathcal{N}(\mathbf{0}, \mathbf{I})$ given $x_0 \sim \mathcal{N}(\mathbf{0}, \mathbf{I})$ ( Equation 2), such that sampling initial noise from $\mathcal{N}(\mathbf{0}, \mathbf{I})$ during inference does not introduce the noise distribution misalignment. In contrast, for data with a large domain shift to LAION (e.g. satellite data) the pre-trained VAE can fail to effectively map input images from these domains to a latent distribution following $\mathcal{N}(\mathbf{0}, \mathbf{I})$. Consequently, $x_0$ and $x_T$ during training do not match a standard Gaussian. This amplifies the noise mismatch between training and inference, making it harder to accurately reconstruct low-frequency features (e.g., color) during inference.

## 4 METHODS

Building on Section 3, we propose two methods to enhance synthetic satellite image generation.

### 4.1 NOISE CALIBRATION IN TRAINING - OFFSET NOISE

We first propose offset noise to adapt the noise distribution during diffusion training while keeping the noise during diffusion inference unchanged, initially introduced to generate very bright or dark images with Stable Diffusion (Guttenberg, 2023). We extend it to our setup to mitigate signal leak bias, enabling the model to capture low-frequency features in out-of-distribution satellite data.

Shown in Figure 2 (a), offset noise adds channel-wise Gaussian offset $\epsilon_c$ to standard Gaussian noise $\epsilon_0$ during training. Instead of Equation 2, the noise at the last diffusion step T during training is:

$$x_T = \sqrt{\bar{\alpha}_T}x_0 + \sqrt{1 - \bar{\alpha}_T}(\epsilon_0 + \epsilon_c), \quad \epsilon_0 \sim \mathcal{N}(\mathbf{0}, \mathbf{I}), \epsilon_c \sim \mathcal{N}(\mathbf{0}, \lambda\Sigma_{\mathbf{c}}) \tag{3}$$

Here $\Sigma_{\mathbf{c}}$ is a block-diagonal covariance matrix with ones within each channel and the offset noise $\epsilon_c$ is then constant over all pixels within the same channel and varies across different channels and the magnitude of injected offset noise can be controlled by $\lambda$. Therefore, it does not affect the pixel-wise perturbation introduced by standard Gaussian noise $\epsilon_0$, preserving the model's ability to learn pixel-wise high-frequency details. Furthermore, per-channel mean of the standard Gaussian noise $\epsilon_0$ has very small variance after averaging over $H \times W$ ($64 \times 64$ for input image in resolution $512 \times 512$) pixels, so that it is unable to perturb the signal leakage containing channel-mean information. In contrast, the channel-wise offset explicitly increases the variance of the per-channel mean, injecting a channel-mean shift with the magnitude controlled by a parameter $\lambda$. Since the channel-wise offset is sampled from a zero-mean Gaussian, the channel-mean shift randomly varies during training, sometimes canceling and sometimes amplifying the signal leakage. This randomness prevents the model from relying on the channel-mean information in the signal leakage, forcing it to learn to reconstruct the low-frequency components of the image from noise. Consequently, during inference, the trained model is able to reconstruct low-frequency information from pure noise.

This method directly works at the onset of the signal leakage, enabling the model to dynamically mitigate the effect of signal leakage during training. The scale parameter $\lambda$ offers dataset-dependent

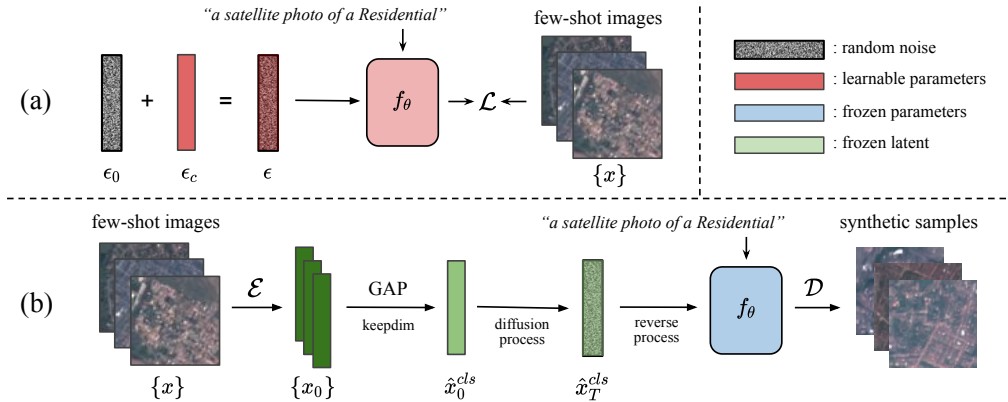

Figure 2: Method overview. (a) Offset noise. $\epsilon_c$ is the channel-wise offset noise providing additional channel mean perturbation. (b) Leak-aligned noise. Global Average pooling (GAP) averages the latent representations of few-shot data over pixels to obtain a chanel-wise mean embedding for each few-shot sample and then average across samples to obtain a class-wise representation.

control over the strength of the channel-mean perturbation. However, it also requires multiple runs of model training to find an optimal offset scale $\lambda$, which demands higher computational resources. Moreover, the internal interaction between offset noise and standard gaussian noise is a black-box.

### 4.2 NOISE CALIBRATION IN INFERENCE - LEAK-ALIGNED NOISE

A simpler approach is to keep training unchanged and calibrate inference noise using the leak-aligned noise method, which uses leakage from the training data to estimate initial noise.

According to Equation 2 presenting the noise during training, we can estimate an initial noise during inference with the computation of $\sqrt{\bar{\alpha}_T}$, $\sqrt{1 - \bar{\alpha}_T}$ and an estimation of $x_0$. Specifically, we use the noise schedule defined in Stable Diffusion (Rombach et al., 2022) to compute $\sqrt{\bar{\alpha}_T}$ and $\sqrt{1 - \bar{\alpha}_T}$ at $T = 1000$. The result shows that $\sqrt{\bar{\alpha}_T} = 0.0682373$ and $\sqrt{1 - \bar{\alpha}_T} = 0.997558$.

As the out-of-distribution input data for fine-tuning the model makes the noise distribution misalignment between model training and inference visible, we estimate $\hat{x}_0$ from the same input few-shot data used for fine-tuning the model. Specifically, as it is reported that the signal leakage term $\sqrt{\bar{\alpha}_T} x_0$ causing noise distribution misalignment contains channel-mean information of the input images, we estimate $\hat{x}_0^{\text{cls}}$ per-class by computing the channel-means of the input few-shot data. To get a similar latent representation as the $x_0$ during training, we use the same VAE encoder as training. Overall, in the inference phase, we replace the initial Gaussian noise with the noise in the following form:

$$\hat{x}_T^{\text{cls}} = \sqrt{\bar{\alpha}_T}\hat{x}_0^{\text{cls}} + \sqrt{1 - \bar{\alpha}_T}\epsilon_0, \quad \epsilon_0 \sim \mathcal{N}(\mathbf{0}, \mathbf{I}) \tag{4}$$

Shown in Figure 2 (b), we provide the trained model $f_\theta$ at inference with the new noise containing signal leakage and thereby channel-mean information similar to that seen during training. The model is able to use this information again during inference to generate images, aligning the channel-mean information in the synthetic image with that in the real image and improving the synthetic quality.

Unlike offset noise, this method requires no additional resources beyond the baseline few-shot fine-tuning. Class-wise estimation of $x_0$ provides class-specific calibration of channel-mean information, capturing class-specific characteristics. However, the robustness of this method against the signal leakage depends on accurately estimating class-wise channel means and knowing the noise schedule.

## 5 EXPERIMENTS

To evaluate our methods on satellite datasets, we run experiments on EuroSAT (Helber et al., 2019), PatternNet (Zhou et al., 2018) and NWPU-RESISC45 (Cheng et al., 2017). For a fair comparison with the baseline DataDream, we use the same 16-shot image set and finetune Stable Diffusion

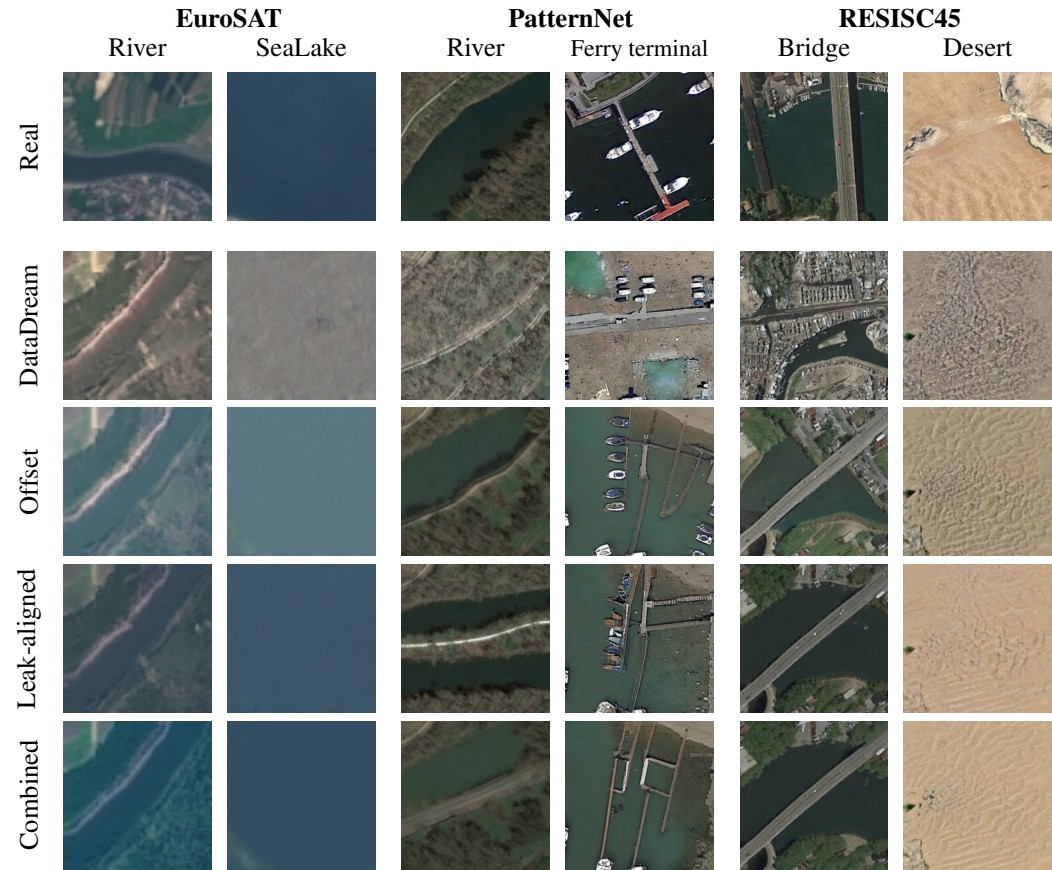

Figure 3: Synthetic image examples from different methods, compared with real samples.

2.1-base (Rombach et al., 2022). We use DataDream's configurations: 200 training epochs, batch size 8, learning rate $= 1e - 4$ for a cosine scheduler. For offset noise, we search for an optimal $\lambda \in [0.01, 0.1]$ to control the amount of offset noise added and find $\lambda = 0.05, \lambda = 0.07, \lambda = 0.035$ give the best results for EuroSAT, PatternNet and NWPU-RESISC45, respectively. For generation, we use 50 inference steps and guidance scale 2.0 to generate 500 images per class. To evaluate the synthetic satellite image quality in adapting the downstream model, we use the images to train a CLIP VIT-B/16 (Radford et al., 2021) classifier. It is trained with LoRA weights on the text and image encoder. Other implementation details are in Appendix A.2.

## 5.1 QUALITATIVE COMPARISON

Figure 3 shows a qualitative comparison of the synthetic images generated by our methods and DataDream, along with the few-shot real images used for fine-tuning. From each benchmark satellite dataset, we select two classes for comparison. Compared to DataDream, our offset noise and leak-aligned noise methods better match the color tones of real satellite images. This is more pronounced in classes dominated by a single color tone, like "SeaLake" in EuroSAT and "desert" in RESISC45. Unlike DataDream's color shifts, our methods correct global color tones and ensure consistency with real few-shot images, especially reproducing water body colors accurately. This is key for improving synthetic satellite image fidelity. Our methods also better separate main features from the background, for example, clearly distinguishing rivers that blend into the background in DataDream images. Combining our methods can enhance color correction, as in EuroSAT's SeaLake class, but may also cause overcorrection, like the greenish tone in the river class.

**Density and Rarity Comparison Between Real and Synthetic Images.** To quantitatively evaluate synthetic data quality and fidelity, we evaluate the distribution shift between real and synthetic datasets with density and rarity. As described in section 3.1, high density and low rarity indicate

Table 2: Comparison of real and synthetic distributions on EuroSAT (EuSAT) (Helber et al., 2019), PatternNet (Patt) (Zhou et al., 2018), and RESISC45 (RESISC) (Cheng et al., 2017).

(a) density Naeem et al. (2020)

|  | EuSAT | Patt | RESISC |
|---|---|---|---|
| DataDream | 0.2189 | 0.0118 | 0.0393 |
| offset noise | 0.3432 | 0.0122 | 0.0527 |
| leak-aligned | 0.3232 | 0.0125 | 0.0528 |
| combined | 0.3552 | 0.0117 | 0.0512 |

(b) rarity Han et al. (2022)

|  | EuSAT | Patt | RESISC |
|---|---|---|---|
| DataDream | 16.3954 | 16.7239 | 16.7930 |
| offset noise | 15.8201 | 16.5441 | 16.3919 |
| leak-aligned | 15.8418 | 16.5586 | 16.8990 |
| combined | 15.8519 | 16.6733 | 16.7942 |

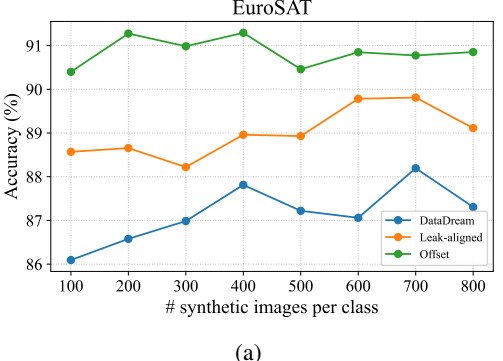

(a)

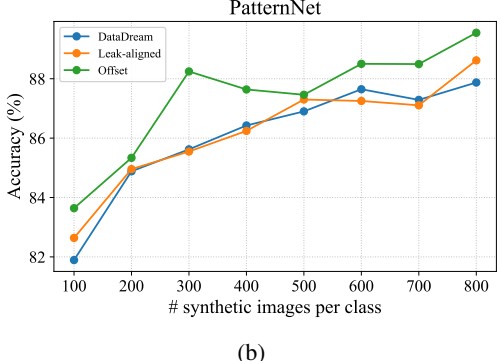

(b)

Figure 4: Effects of increasing the number of synthetic training images per class on CLIP accuracy.

smaller distribution shift and higher image fidelity. In Table 2, we show that synthetic images generated by both offset noise and leak-aligned noise exhibit higher density and lower rarity compared to those generated by DataDream. This demonstrates that synthetic images generated by our methods construct a data distribution closer to the real distribution and exhibit higher fidelity.

## 5.2 QUANTITATIVE COMPARISON

Table 3a shows 16-shot classification results on three satellite datasets using classifiers adapted with synthetic or few-shot real data, averaged over three seeds. Synthetic data from our methods achieves significantly higher top-1 accuracy than other methods. For example, offset noise achieves $90.46\%$ on EuroSAT, resulting a SOTA performance with a $3.24$ percentage point improvement over DataDream. Leak-aligned noise also demonstrates a $1.71$ percentage point improvement over DataDream. On NWPU-RESISC45, leak-aligned noise performs best among all compared methods with $70\%$ top-1 accuracy. On PatternNet, offset noise and leak-aligned noise perform comparably, with $0.56$ and $0.4$ percentage point improvement over DataDream.

The impact of combining methods varies by dataset: for EuroSAT, it performs between offset noise and leak-aligned alone; for PatternNet, it slightly outperforms both; and for RESISC, it underperforms both individually. We attribute this to overcorrection of noise distribution misalignment when combining offset and leak-aligned noise, along with varying sensitivity of different satellite datasets to such overcorrection. As explained in Section 4, offset noise and leak-aligned noise address noise misalignment from complementary angles. Offset noise perturbs per-channel mean leakage, aligning the training noise distribution closer to a standard Gaussian and reducing misalignment during inference. In contrast, leak-aligned noise addresses misalignment by adjusting inference noise to match training noise with signal leakage. Their differing assumptions about the target noise can cause overcorrection when combined, with varying effects across satellite datasets. For datasets like PatternNet, the main characteristic (described in class name) is clear and easy to be distinguished between categories, such that the overcorrection of channel mean information is acceptable and has less impact on the classification performance. For datasets like RESISC45, the scene complexity of the satellite images is increased, requiring fine-grained classification. Consequently, classification is more sensitive to slight channel mean overcorrection, reducing performance.

Table 3: Accuracy of CLIP ViT-B/16 fine-tuned on 500 synthetic images per class generated with 16 real examples. Real fine-tuned refers to training the base classifier with 16 real few-shot images.

(a) On satellite datasets, offset and leak-aligned noise improve performance.

|  | EuroSAT | RESISC45 | PatternNet |
|---|---|---|---|
| zero-shot | 40.26 | 27.26 | 34.01 |
| DataDream (Kim et al., 2024) | $87.22_{\pm 1.52}$ | $67.94_{\pm 0.91}$ | $86.90_{\pm 1.24}$ |
| offset noise | $\mathbf{90.46}_{\pm 1.59}$ | $\mathbf{68.94}_{\pm 0.22}$ | $\mathbf{87.46}_{\pm 1.68}$ |
| leak-aligned noise | $\mathbf{88.93}_{\pm 1.68}$ | $\mathbf{70.00}_{\pm 0.58}$ | $\mathbf{87.30}_{\pm 0.91}$ |
| Combined | $\mathbf{89.47}_{\pm 0.61}$ | $68.22_{\pm 1.01}$ | $\mathbf{88.46}_{\pm 1.11}$ |
| real fine-tuned | $92.60_{\pm 0.41}$ | $82.61_{\pm 0.55}$ | $94.91_{\pm 0.53}$ |

(b) For in-distribution datasets, DataDream suffices without our methods.

|  | Oxford Pets | Caltech101 | Food101 |
|---|---|---|---|
| DataDream (Kim et al., 2024) | 93.33 | 96.52 | 86.43 |
| leak-aligned noise | 93.30 | 96.64 | 86.68 |
| real fine-tuned | 93.45 | 97.25 | 87.30 |

(c) Classification performance of offset noise at varying offset scales.

|  | 0.03 | 0.04 | 0.05 | 0.06 | 0.07 | 0.08 |
|---|---|---|---|---|---|---|
| EuroSAT | 89.61 | 89.88 | 91.20 | 88.18 | 85.71 | 86.33 |
| PatternNet | 87.04 | 87.74 | 88.09 | 87.60 | 89.39 | 86.74 |
| RESISC | 68.59 | 67.91 | 67.34 | 66.21 | 66.39 | 63.93 |

## 5.3 ABLATIONS

**Number of synthetic images.** We examine if classifier performance improves with more synthetic data. Figure 4 shows that on EuroSAT, offset noise and leak-aligned noise are less sensitive to the number of synthetic images, while DataDream's performance slightly improves as the number of synthetic images scales up. On PatternNet, model performance improves with more synthetic images from all three methods.

**Effectiveness on In-distribution datasets.** In addition to datasets with large distribution shift to LAION, we apply leak-aligned noise to in-LAION-distribution datasets Oxford Pets (Parkhi et al., 2012), Caletch101 (Li et al., 2022) and Food101 (Bossard et al., 2014), to emphasize that our methods are specifically designed for out-of-LAION-distribution datasets like satellite data. In Table 3b, leak-aligned noise performs comparably to the SOTA method DataDream on in-distribution datasets.

**Effect of Offset scale** We vary the offset noise scale from 0.03 to 0.08 in 0.01 increments to assess performance sensitivity. Table 3c shows that while offset scale impacts performance, the effect is not monotonic. Hence, we pick the offset scales that peaks the model performance for each dataset.

## 6 CONCLUSION

We investigate how distribution shifts between target images and synthetic training data limit few-shot fine-tuning and degrade synthetic quality. Our analysis reveals a strong link between distribution shift and the performance gap in synthetic-versus-real classification, which stems from misaligned noise distributions in diffusion models caused by signal leakage during training. To address this, we propose two methods: offset noise, which reduces leakage during training, and leak-aligned noise, which calibrates inference noise to match training conditions. Experiments on satellite data demonstrate that both methods improve synthetic quality and classifier performance.

## 7 REPRODUCIBILITY STATEMENT

To ensure reproducibility, we run all experiments with three seeds.

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

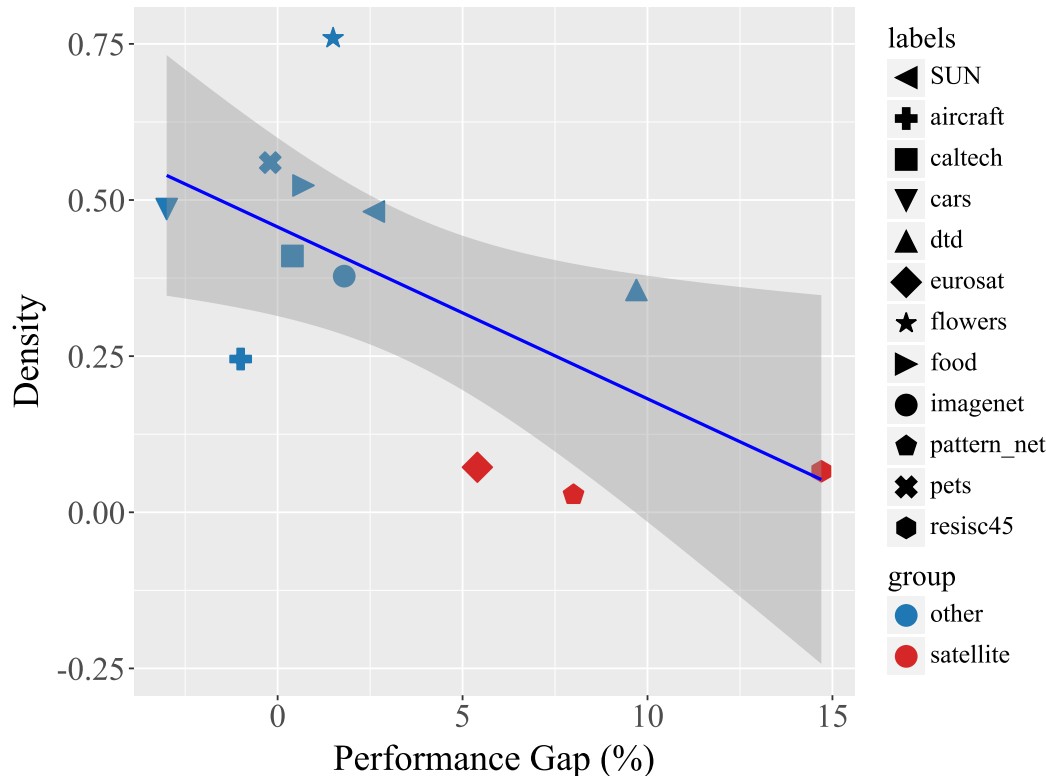

Figure 5: Correlation between density and performance gap.

## A APPENDIX

### A.1 USE OF LLMS

This work used LLMs to improve sentence clarity and flow.

### A.2 IMPLEMENTATION DETAILS

For the few-shot fine-tuning of text-to-image model Stable Diffusion (Rombach et al., 2022), we use the standard CLIP prompt specialized for satellite imagery as text condition: "a satellite photo of a [CLS]", where [CLS] refers to a classname. Fine-tuning is achieved by adding LoRA adaptors to the attention layers of both the text encoder and the U-Net. We train the LoRA weights with rank $r = 16$ and $\alpha = 16$. For image generation, we use the same prompt template as fine-tuning.

For classifier training, we add LoRA adaptors to the text encoder and image encoder of CLIP VIT-B/16 model (Radford et al., 2021) and set $r = 16$ and $\alpha = 32$. In addition, we use data augmentation strategies such as random horizontal flip, random color jitter, random gray scale, Gaussian blur and solarization. Specifically, we apply random resized crop and set the lower bound $= 0.8$ and the upper bound $= 1.0$ for the random area of the crop before resizing (i.e. scale $= (0.8, 1.0)$).

### A.3 ANALYSIS RESULTS

In Figure 5, we present the correlation plot between density and classification performance gap. According to the statistics Pearson coefficient $R = -0.63$ and p-value $= 0.0283 < 0.05$, we confirm the significance of the correlation. It implies that large distribution shift (low density) is significantly correlated to the classification performance gap between classifiers training with solely synthetic and solely real few-shot data.

