# OpenReview forum: "Aligning Signal Leakage Matters for Synthetic Data Generation of Satellite Imagery"
_ICLR.cc/2026/Conference — ICLR 2026 Conference Withdrawn Submission_

### Official Review · Reviewer_aw5K · 2025-10-31

**Soundness:** 2
**Presentation:** 2
**Contribution:** 2
**Rating:** 4
**Confidence:** 3

**Summary:**

This work studies the distributional gap of the pretraining datasets used to train large diffusion models and out-of-distribution datasets, such as those in the remote sensing domain. Particularly, they focus attention on synthetic data generation for augmenting training datasets in the remote sensing domain. Authors investigate leakages in current diffusion models that hinder their finetuning for custom downstream datasets. Their main finding is addressing _offset noise_ and _leak-aligned noise_ in the diffusion process. Noise adaptations are evaluted with downstream model performance.

**Strengths:**

- Authors shown an important observation: noise schedules in diffusion models don't reach pure Gaussian noise, thus leaking low-frequency biases during training. These biases are then reflected at sampling time when using pure Gaussian noise.
- Authors carry out an extensive analysis on the identified limitation of current diffusion models, and propose two possible methods for mitigating such bias in satellite images. Importantly, one of the two proposed methods doesn't require retraining of the base diffusion models, rather is an adaptation at inference time.
- Authors justify claims with theoretical intuitions and empirical evidence.
- Overall, I believe this is an interesting research direction that is overlooked in the standard image domains.

**Weaknesses:**

- Rarity and density metrics are determined empirically and with little datasets. For instance, authors claim that _rarity_ metric is better simply by looking at DTD dataset and seeing that it reflects better the OOD shift. Base this decision on just one dataset is not enough.
- Satellite data augmentation in the RGB domain is a rather simplified setting. Satellite images operate on a non-normalised rgb spectrum, having more non-visible channels.
- Why is the "combined" setting not plotted in Figure 4?
- The difference in the results obtained for PatternNet dataset do not look significant enough. Do authors have any intuition about this?
- Experiments seem rather limited, just evaluating on 3 datasets. As mentioned above, the remote sensing domain would benefit if such methods were also adapted for not only RGB images but rather supporting full-spectrum satellite image generation.

**Questions:**

- Why do authors limit their evaluations on remote sensing images? Other OOD settings such as X-ray scans, or other "rare" domains could have been tested to further prove the effectiveness of the method proposed.

---

### Official Review · Reviewer_nWj6 · 2025-10-31

**Soundness:** 2
**Presentation:** 2
**Contribution:** 1
**Rating:** 2
**Confidence:** 4

**Summary:**

The authors observed that synthetic data generated from Stable Diffusion does not perform as well for domains outside the model's training.
Inspired by recent work that showed how the initial noise plays a significant role in the diffusion inference process, they propose a lightweight method (on top of a low-rank adaptation fine-tuning approach) to improve the synthetic data quality for these domains by altering the initial noise used in generating an image. In their experiments, they show that the proposed algorithm can improve the classification accuracy of a classifier trained only on synthetic images.

**Strengths:**

- The paper sets up the problem well, with the authors first showing how synthetic data for domains outside the training distribution of Stable Diffusion performs worse (Figure 1) and then trying to measure this difference between different datasets and LAION using DINOv2 representations. This motivates their approach for improving the synthetic data quality for a downstream task.

- The idea of tuning the initial noise in the diffusion process to improve the domain-specific generation quality of the model is interesting and a novel approach that is orthogonal to existing fine-tuning methods.

**Weaknesses:**

- The motivation for why signal leakage has a stronger impact on non-LAION datasets is flawed. The authors claim that "*Diffusion’s VAE maps LAION-like data to a standard Gaussian in latent space*" (lines 233-234). This is far from true, as the KL term in the VAE training has little weight and, in practice, any image encoded with the trained VAE is probably far from the unit Gaussian. Additionally, if this were true, the authors should have validated it with a simple experiment of mapping different images to the latent space and measuring the KL divergence.

- The results are limited to three datasets, questioning the generalizability of the method. In one of the three datasets (RESISC45), the proposed method does not improve the downstream classification, and in all cases, the reported gains are minimal. Overall, the experiments do not convince the reader that leakage of the initial noise is a huge issue, since the base LoRA-finetuned model performs almost as well.

- The experiments are limited to the satellite domain. Is this domain really that far out of the distribution of the Stable Diffusion training data? The authors should have tried another domain where this difference is more pronounced, e.g., medical images (histopathology, MRI), and potentially show that, there, the leakage has a significant impact on the synthetic data generation.

**Questions:**

- How does the method scale when there's more data to train the LoRA with? One would assume that if the LoRA has enough data, this issue of the leakage effect of the initial noise could be mitigated. Are there cases where the LoRA cannot fix it, and your method is necessary?

---

### Official Review · Reviewer_r27e · 2025-11-01

**Soundness:** 2
**Presentation:** 3
**Contribution:** 2
**Rating:** 4
**Confidence:** 4

**Summary:**

This paper addresses the challenge of generating synthetic training data for satellite imagery using diffusion models, particularly when there is a significant distribution shift from the pre-training data (e.g., LAION). The authors identify that data rarity strongly correlates with the performance gap between models trained on real vs. synthetic data. To mitigate this, they propose two noise calibration methods—offset noise and leak-aligned noise—designed to align low-frequency characteristics (e.g., color) in synthetic satellite images. Experiments on three satellite benchmarks show improvements over the DataDream baseline in both qualitative and quantitative evaluations.

**Strengths:**

1.	The analysis of how the target data distribution affects synthetic data quality is novel, and the use of rarity and density metrics to quantify this distribution shift is well-motivated.
2.	This paper proposes two simple and complementary improvements: offset noise (training correction) and leak-aligned noise (inference correction). Both methods are lightweight and easily implementable in LoRA/fine-tuning workflows.
3.	The method was evaluated on three satellite datasets (EuroSAT, PatternNet, and NWPU-RESISC45), where it exhibited performance superior to that of DataDream.

**Weaknesses:**

1. The proposed method differs from DataDream only in its handling of the noise distribution. Since "offset noise" is a technique previously introduced by Guttenberg (2023), the core contribution of this work is merely its application to alleviate signal leakage. This constitutes a valid but minor advancement.
2. The results in Table 3(a) reveal a lack of a consistent pattern in the performance of the proposed methods. The fact that each technique—Offset Noise, Leak-Aligned Noise, and their combination—excels on a different dataset (EuroSAT, RESISC45, and PatternNet, respectively) fails to clarify their synergistic relationship or the principles for selecting one over the other.
3. As reported in Table 3(c), the classification performance of offset noise varies significantly with different offset scales across datasets. For instance, the optimal offset scale is 0.05 for EuroSAT, 0.07 for PatternNet, and 0.03 for RESISC45. This indicates that extensive experiments are required to tune the offset scale for each dataset to achieve the best performance. The selection of the offset scale substantially impacts the experimental results. For example, on the PatternNet dataset, the accuracy is 87.71 when the offset scale is 0.07, which is lower than the baseline of 87.22. However, when the offset scale is 0.05, the reported accuracy reaches 91.20—a result that appears to be higher than the corresponding value reported in Table 3(a).
4. From the visual results presented in Figure 3, the method proposed in this paper appears to generate images with better color fidelity to the real samples compared to the DataDream approach. However, it may still struggle with generating fine-grained details. For instance, in the "Ferry terminal" scene, the shapes of the synthesized boats appear distorted, and the bridge structure is unnaturally curved.

**Questions:**

see weaknesses.

---

### Official Review · Reviewer_U8FA · 2025-11-03

**Soundness:** 2
**Presentation:** 3
**Contribution:** 3
**Rating:** 4
**Confidence:** 2

**Summary:**

The paper studies when synthetic data from diffusion models falls short when the target data distribution has a large distribution shift from the model's pretraining data distribution. Such distributional shift is specifically significant for satellite images which are rare and causes more significant noise misalignment in diffusion model's training and inference stages. Two methods are proposed to mitigate the misalignment and enhance the quality of synthetic satellite images.

**Strengths:**

1. The paper is generally clearly written.

2. The paper aims to analyze and establish the correlation between pretraining/target data distribution shift and the effectiveness of using corresponding synthetic data in training downstream models. Empirical results support this intuition.

3. The proposed methods mitigate the noise misalignment problem. Qualitatively, the satellite images generated has higher quality (more faithful color) and yields better performances when used to train downstream models.

**Weaknesses:**

1. Experiments in Section 3 does support the claim that larger distribution shift in training/target data distribution correlates with the effectiveness of using synthetic data in training downstream models. However, there seems to lack experimental supports for why noise misalignment may be more severe for data with larger distributional shift. Specifically, is there empirical results to validate the hypothesis mentioned in line 230-240?

2. It seems to me that the offset noise method is a method that was used in prior work for slightly different purposes. It is thus not clear to me the technical contribution of this work. Could the authors clarify how the proposed methods different from existing techniques?

**Questions:**

Please see above. My main concern is about empirical results validating the connection between noise misalignment and data distributional shift, as well as the originality of the proposed methods. I'd be open to adjust my score given the author's response.

---

### Note · Authors · 2026-03-12

I have read and agree with the venue's withdrawal policy on behalf of myself and my co-authors.

---

### Meta-Review · Area_Chair_FDeA · 2025-12-08

**Summary:**

This paper was reviewed by 4 knowledgeable reviewers whose main concerns were:
- The novelty of the proposed approach, which appeared to repurpose existing work to avoid signal leakage (U8FA, r27e).
- The unclear significance of the presented results (U8FA, aw5K) and how well the experiments support the claims (r27e, nWj6).
- The validity of the claim on signal leakage and its impact on non-LAION datasets (nWj6).

**Reviewer Concerns:**

There was no rebuttal so all concerns remain.

**Reviewer Scores:**

There was no rebuttal, so no impact on the discussion/scores.

---

### Decision · Program_Chairs · 2026-01-26

Reject